# Antithrombotic PreTreatment and Invasive Strategies in Patients with Non-ST-Segment Elevation Acute Coronary Syndrome

**DOI:** 10.3390/jcm9082578

**Published:** 2020-08-09

**Authors:** Cyril Camaro, Peter Damman

**Affiliations:** Department of Cardiology, Radboud University Medical Center, P.O. Box 9101, 6500 HB Nijmegen, The Netherlands; cyril.camaro@radboudumc.nl

**Keywords:** non-ST segment elevation acute coronary syndrome, antithrombotic treatment, routine invasive strategy, optimal timing

## Abstract

In the current era, the antithrombotic treatment of patients with non-ST segment elevation acute coronary syndrome (NSTE-ACS) includes standard aspirin, and one of the potent P2Y_12_ inhibitors ticagrelor or prasugrel. The optimal timing of ticagrelor has not been adequately studied, while prasugrel is only recommended after coronary angiography prior to PCI. The invasive strategy, including indication and timing of angiography, depends on risk stratification and a mortality benefit has been shown in selected high-risk NSTE-ACS undergoing early (<24 h) intervention.

## 1. Introduction

The acute treatment of patients diagnosed with non-ST segment elevation acute coronary syndrome (NSTE-ACS), consists of pharmacological treatment and potential invasive coronary angiography with subsequent revascularization if necessary. Whereas, in ST-segment elevation myocardial infarction, characterized by an occluded epicardial coronary artery, primary percutaneous intervention (PCI) as soon as possible results in improved outcomes, the invasive treatment decision in NSTE-ACS is more complex. In NSTE-ACS typically angiographically significant stenosis is observed, while the epicardial flow is maintained. In this review, we focus on antithrombotic treatment strategies (before or at the time of coronary angiography) and the invasive management in NSTE-ACS.

## 2. Antithrombotic Pretreatment Strategies

Antithrombotic treatment (ATT) is the cornerstone in NSTE-ACS patients regardless of invasive management. The choice, combination, timing of initiation and treatment duration, all depend on patient characteristics, including the ischemic and bleeding risk, as well as on procedural aspects. Activation of platelet aggregation and the coagulation cascade plays a key role in the initial phase and evolution of NSTE-ACS. Therefore, platelet inhibition and anticoagulation are important, especially in those patients undergoing revascularization. 

### 2.1. Oral Antiplatelet Therapy

Aspirin or acetylsalicylic acid (ASA) is the basis of antiplatelet inhibition. The clopidogrel in unstable angina to prevent recurrent ischemic Events (CURE) trial showed that the addition of clopidogrel results in improved clinical outcomes [1]. Based on the results of the PLATO (platelet inhibition and patient outcomes) and TRITON-TIMI 38 (trial to assess improvement in therapeutic outcomes by optimizing platelet inhibition with prasugrel–thrombolysis in myocardial infarction 38) trials, where the superiority of the novel P2Y_12_ receptor inhibitors has been shown over clopidogrel, dual antiplatelet treatment (DAPT), including ASA and a potent P2Y_12_ receptor inhibitor (ticagrelor or prasugrel), are recommended by current clinical guidelines [2,3]. 

While ASA is started as soon as possible, the optimal timing of initiation of the P2Y_12_ receptor inhibitor is a matter of debate. Pretreatment with P2Y_12_ inhibition includes a variety of different scenarios. The P2Y_12_ inhibitor can be given prehospital (ambulance), at the emergency department or even in the catheterization laboratory after coronary angiography (CAG), but before PCI. The rationale for pretreatment in NSTE-ACS is to achieve sufficient platelet inhibition at the time of PCI, avoiding ischemic complications, such as acute stent thrombosis. Disadvantages are a high bleeding risk in patients with extensive coronary artery disease with an indication for revascularization by urgent coronary artery bypass grafting (CABG), or those who turn out to have another diagnosis, such as aortic dissection. The pretreatment with clopidogrel the Antiplatelet Therapy for Reduction of Myocardial Damage During Angioplasty (ARMYDA)-5 PRELOAD study randomized 409 patients to receive 600 mg clopidogrel loading dose 4–8 h before PCI or a 600 mg loading dose in the catheterization laboratory after coronary angiography, but before PCI [4]. In this trial no significant difference in the 30-day incidence of major adverse cardiac events (MACE): A composite of cardiac death, myocardial infarction, or unplanned target vessel revascularization, between the pretreatment and no pretreatment groups (8.8% vs. 10.3%; *p* = 0.72). No increased risk of bleeding or vascular complications with pretreatment was observed either (5.4% vs. 7.8%; *p* = 0.42). The main limitation of this study is the small proportion of patients with NSTE-ACS (only 39% of the study population). Randomized pretreatment studies with ticagrelor in NSTE-ACS have not been performed. However, in the PLATO trial, 18,624 patients were pretreated with ticagrelor versus clopidogrel before undergoing coronary angiography. Ticagrelor reduced the risk of ischemic events as compared to clopidogrel (9.8 vs. 11.7%. hazard ratio (HR), 0.84; 95% confidence interval [CI], 0.77 to 0.92; *p* < 0.001) [2]. Prasugrel has been investigated more frequently than ticagrelor. In the TRITON-TIIMI 38 trial, 13608 patients with moderate-to-high-risk acute coronary syndromes (ACS) were randomly allocated to prasugrel or clopidogrel. All patients randomly allocated to prasugrel were given 60 mg after coronary angiography, when the anatomy was known. This study showed significant reductions in the prasugrel group in the rates of myocardial infarction (9.7% for clopidogrel vs. 7.4% for prasugrel), urgent target vessel revascularization (3.7% vs. 2.5%) and stent thrombosis (2.4% vs. 1.1%). In the ACCOAST trial (ACCOAST—comparison of prasugrel at the time of percutaneous coronary intervention or as pretreatment at the time of diagnosis in patients with non-st elevation myocardial infarction), patients were randomized to prasugrel pretreatment or a strategy comparable to TRITON-TIMI 38 [5]. In the ACCOAST trial, 4033 patients with NSTE-ACS and a positive troponin level scheduled for CAG were randomized to prasugrel 30 mg before CAG (pretreatment group) or placebo (control group). When CAG was followed by PCI, an extra 30 mg of prasugrel was added in the pretreatment group at the time of PCI, while control patients receive the complete 60 mg of prasugrel. No reduction in the primary endpoint (composite of cardiovascular death, recurrent myocardial infarction, stroke, urgent revascularization or glycoprotein IIb/IIIa rescue therapy through day 7) was found with pretreatment of 30 mg prasugrel (HR with pretreatment 1.02; 95% CI, 0.84–1.25; *p* = 0.81). There was, however, a significant increase in thrombolysis in myocardial infarction (TIMI) major bleeding, whether or not related to CABG, at day 7 in the pretreatment group (2.6% vs. 1.4%. HR 1.90; 95% CI 1.19–3.02; *p* = 0.006). This trial was prematurely interrupted on recommendation from the data safety monitoring board when 398 of the 400 intended primary endpoint events had been collected, corresponding to 4033 of the approximately 4100 patients originally planned. The increase in TIMI major bleeding was irrespective whether the procedure was performed by radial or femoral access [6]. Therefore, there is inconclusive evidence regarding the optimal timing of P2Y_12_ inhibition. 

With regard to the optimal P2Y_12_ inhibitor, the two potent P2Y_12_ inhibitors ticagrelor and prasugrel were compared head-to-head in the ISAR-REACT 5 trial [7]. In this multicenter, randomized, open-label trial, 4018 patients with ACS (58.9% were NSTE-ACS) were randomly assigned to ticagrelor (180 mg loading dose before coronary angiography) or prasugrel (60 mg loading dose at the time of PCI). The primary endpoint (a composite of death, myocardial infarction, or stroke at one year) occurred in 184 of 2012 patients (9.3%) in the ticagrelor group and in 137 of 2006 patients (6.9%) in the prasugrel group (HR, 1.36; 95% CI, 1.09–1.70; *p* = 0.006. There was no significant difference in bleeding. The ISAR-REACT 5 trial results have to be interpreted with caution. Around a third of patients were not treated with the assigned drug. Furthermore, the incidence of the primary outcome in ticagrelor-treated patients in ISAR-REACT 5 and PLATO was similar at 9.3% and 9.8%. However, the comparison of the results of prasugrel-treated patients in ISAR-REACT 5 and TRITON-TIMI 38 study shows a large difference: The incidence of the primary outcome being 6.9% and 9.9%, respectively. Despite these limitations, this is the only randomised clinical trial comparing these drugs, and further data is necessary to determine whether prasugrel is the superior P2Y_12_ inhibitor. 

### 2.2. Current Guideline Recommendation

In the 2015 European Society of Cardiology (ESC) Guidelines for the management of acute coronary syndromes in patients presenting without persistent ST-segment elevation no recommendation for or against pretreatment with these agents can be formulated [8]. Prasugrel in whom coronary anatomy is not known is not recommended.

### 2.3. Intravenous Antiplatelet Therapy

The P2Y_12_ inhibitor cangrelor, an i.v. adenosine triphosphate (ATP) analogue that binds reversibly and with high affinity to the P2Y_12_ receptor, and has a fast onset and fast offset of action. This might provide a desirable combination of protection from ischemia without an excessive risk of bleeding. In the placebo-controlled CHAMPION-PLATFORM (Cangrelor Versus Standard Therapy to Achieve Optimal Management of Platelet Inhibition) trial, 5362 patients with NSTE-ACS were randomly assigned to cangrelor or placebo at the time of PCI, followed by 600 mg clopidogrel [9]. No reduction in the primary endpoint (a composite of death, myocardial infarction, or ischemia-driven revascularization at 48 h) was found in the cangrelor group, as compared with the placebo group (odds ratio (OR) in the cangrelor group, 0.87; 95% CI, 0.71–1.07; *p* = 0.17). In CHAMPION-PHOENIX, 11,145 patients who were undergoing either urgent or elective PCI and were randomized to a bolus and infusion of cangrelor or to receive a loading dose of 600 mg or 300 mg of clopidogrel. The rate of the primary efficacy endpoint (a composite of death, myocardial infarction, ischemia-driven revascularization, or stent thrombosis at 48 h) was 4.7% in the cangrelor group and 5.9% in the clopidogrel group (*p* = 0.005). Stent thrombosis occurred in 0.8% of the patients receiving cangrelor and in 1.4% in the clopidogrel group (OR, 0.62; 95% CI, 0.43–0.90; *p* = 0.01). There was no significant increase in bleeding. With its ability to reduce ischemic complications of PCI, cangrelor can be implemented in situations in which ADP-receptor blockade is needed, but a short-acting intravenous agent would be preferred, such as patients waiting to undergo CABG.

The use of Intravenous glycoprotein (GP) Iib/IIIa inhibitors as pretreatment have been extensively studied, and attention should be given to the Acute Catheterization and Urgent Intervention Triage Strategy (ACUITY) timing trial [10]. In this trial, 9207 patients with moderate to high-risk ACS were randomly assigned to routine GPIIb/IIIa inhibitor treatment (eptifibatide or abciximab) before PCI or deferred selective use. The primary endpoint (a composite of death, myocardial infarction or unplanned revascularisation for ischemia) at 30 days occurred in 7.1% in the routine group and 7.9% in the deferred group (relative risk, 1.12; 95% CI, 0.97–1.20; *p* = 0.044 for noninferiority; *p* = 0.13 for superiority). There was less major bleeding in the deferred group compared to routine administration at 30 days (4.9% vs. 6.1%; *p* < 0.001 for non-inferiority; *p* = 0.009 for superiority). Eptifibatide was also studied in the EARLY ACS (Early versus Delayed, Provisional Eptifibatide in Acute Coronary Syndromes) trial [11]. Early administration of this GPIIb/IIIa inhibitor leads to an increased risk of major bleeding, while ischemic events did not meet statistical significance. 

### 2.4. Current Guideline Recommendation

Based on these trials, the current ESC guideline states that it is not recommended to administer GPIIb/IIIa inhibitors in patients in whom coronary anatomy is not known. The use of GPIIb/IIIa is reserved for the bailout setting with thrombotic complications.

### 2.5. Anticoagulants During the Acute Phase

In addition to antiplatelet therapy, anticoagulants are proven to be effective in reducing ischemic events in NSTE-ACS. With regards to the acute phase, the use of anticoagulant therapy is an essential adjunct to antiplatelet therapy in the acute treatment of ACS, and is limited to treatment during initial hospitalization and revascularization. In a meta-analysis of 12 trials involving over 17.000 patients, short-term unfractionated heparin, or low molecular weight heparin halved the risk of myocardial infarction or death up to 7 days [12]. Based on these trials, in which therapy was started within a few hours after diagnosis, it is recommended that this therapy is started early. No trials have been performed investigating the exact timing of anticoagulants. Anticoagulants are generally given until revascularization or discharge. 

### 2.6. Unfractioned Heparin (UFH) and Low Molecular Weight Heparin (LMWH).

As described above, the use of UFH or LMWH in addition to aspirin has been shown to be associated with lower short-term ischemic events. UFH is usually weight-adjusted, and monitored using activated clotting times. LMWH, the most widely used being enoxaparin, is dosed based on the weight. Based on a more dose-effect relationship that is more predictable, no monitoring is necessary. UFH has been compared with LMWH, and meta-analyses have shown less ischemic and bleeding outcomes with the use of LMWH [13,14]. However, although these meta-analyses include patients with NSTE-ACS, dedicated NSTE-ACS trials are lacking. 

### 2.7. Fondaparinux

Fondaparinux is a synthetic pentasaccharide that reversibly binds to antithrombin. Dosing is not based on weight, and monitoring is not necessary, and fondaparinux does not lead to heparin-induced thrombocytopenia. In the (Organisation to Assess Strategies in Acute Ischemic Syndromes) OASIS-5 trial, fondaparinux and enoxaparin were compared. Fondaparinux was non-inferior to enoxaparin, but halved in hospital major bleeds at follow-up at nine days [15]. Furthermore, there was a significant mortality benefit at six-month follow-up. This mortality benefit was not reproduced in a large Swedish registry [16]. Despite the superior safety profile, fondaparinux was associated with more catheter thrombosis than enoxaparin, requiring the additional administration of UFH at the time of PCI.

### 2.8. Bivalirudin

Bivalirudin directly binds to thrombin. Bivalirudin has been compared to UFH in the ACUITY trial [17]. Patients were randomized to UFH/LMWH plus GPIIb/IIIa inhibition, bivalirudin plus GPIIb/IIIa inhibition or bivalirudin with bailout use GPIIb/IIIa inhibition. In this trial, while ischemic outcomes were comparable, bivalirudin with bailout use GPIIb/IIIa inhibition resulted in significantly less bleeding. However, the reduction in bleeding seems to be explained by concomitant GPIIb/IIIa inhibition. In the MATRIX (minimizing adverse hemorrhagic events by transradial access site and systemic implementation of angiox) trial, with a more selective GPIIb/IIIa inhibitor use in the UFH arm, no difference in a combination of ischemic and bleeding outcomes was observed [18]. With regards to individual outcomes, stent thrombosis was increased, and bleeding risk decreased with bivalirudin. In VALIDATE-SWEDEHEART (Swedish web-system for enhancement and development of evidence-based care in heart disease evaluated according to recommended therapies) trial, UFH use was compared with bivalirudin with limited use of GPIIb/IIIa inhibitors [19]. The investigators showed comparable risks for both ischemia and bleeding when comparing the two drugs.

### 2.9. Anticoagulant and Treatment Strategy

Based on the above, in invasively managed patients, UFH is the standard of choice. Bivalirudin can be used as an alternative to UFH. In conservatively managed patients, fondaparinux is the first, choice. In case of cross-over from a conservative to an invasive strategy, additional UFH or bivalirudin is necessary for PCI. 

### 2.10. Current Guideline Recommendation

In the 2015 ESC guideline, UFH and bivalirudin are recommended in NSTE-ACS undergoing PCI, while fondaparinux (with additional UFH during PCI) is recommended regardless of the treatment strategy. Enoxaparin is recommended if fondaparinux is not available. 

## 3. Invasive Coronary Angiography and Revascularization

CAG has an important role in NSTE-ACS. It can confirm or rule out (obstructive) coronary artery disease, and as a consequence, guide antithrombotic therapy. Second, the culprit and non-culprit lesion(s) can be identified; establishing the indication for coronary revascularization by either PCI or CABG. Finally, it can assist in prognostication by assessing the patient’s risk on the basis of coronary anatomy.

Comparable to STEMI, revascularization in selected NSTE-ACS patients improves 1-year survival [20]. However, different treatment strategies have emerged over the last two decades. They differ with regard to the decision for invasive management and the timing of invasive management. The decision for an invasive strategy should carefully weigh the patient risk profile and coronary angiography/revascularisation-related risk. These include numerous factors, such as clinical presentation, comorbidities, risk scores, the occurrence of high-risk characteristics specific for either CABG or PCI, frailty, cognitive status, estimated life expectancy and the complexity of coronary anatomy. 

### 3.1. Invasive Versus Conservative Treatment

In general, patients admitted with NSTE-ACS undergo either scheduled for routine CAG and revascularization or a selective (conservative) approach. The routine invasive strategy consists of early CAG within 24 to 72 h after admission and subsequent PCI or CABG, where appropriate. The selective invasive or conservative strategy consists of initial pharmacological therapy and only CAG with additional PCI or CABG if the pharmacological therapy fails and/or (recurrent) ischemia is detected. Multiple clinical trials have been performed comparing these two treatment strategies, and current guideline recommendations are mainly based on the following three trials with long-term follow-up. 

In the ICTUS (Invasive Versus Conservative Treatment in Unstable Coronary Syndromes) trial 1200 patients with NSTE-ACS, including an elevated cardiac troponin T, were randomly assigned to an early invasive strategy or to a more conservative (selectively invasive) strategy. No benefit was seen for the early invasive strategy in terms of death, non-fatal myocardial infarction, or rehospitalization for anginal symptoms within one year after randomization [21]. The FRISC-II (Fragmin and Fast Revascularization during Instability in Coronary Artery Disease) investigates 2457 patients with NSTE-ACS who were randomly allocated to invasive or non-invasive treatment [22]. The composite of death or myocardial infarction occurred in 127 (10.4%) versus 174 (14.1%) patients (risk ratio (RR) 0.74 [0.60–0.92], *p* = 0.005). There were also reductions in readmission (451 [37%] vs. 704 [57%]; RR 0.67 [0.62–0.72], *p* < 0.001), and revascularization after the initial admission (92 [7.5%] vs. 383 [31%]; RR 0.24 [0.20–0.30], *p* < 0.001). Moreover, the RITA-3 (randomized trial of a conservative treatment strategy versus an interventional treatment strategy in patients with unstable angina) randomized 1810 patients with NSTE-ACS to early intervention or conservative treatment [23]. In this study, an invasive approach was a better strategy, but the difference was mainly driven by a reduction in refractory angina pectoris. The primary endpoint (a combined rate of death, non-fatal myocardial infarction, or refractory angina at four months; and a combined rate of death or non-fatal myocardial infarction at one year) was met in 86 (9.6%) of 895 patients in the intervention group, compared with 133 (14.5%) of 915 patients in the conservative group (RR 0.66, 95% CI 0.51–0.85, *p* = 0.001). While the early benefit of a routine invasive strategy has, thus, been shown, long-term results are varying. Therefore, Fox et al. performed a pooled analysis of all randomized studies (FRISC-II, ICTUS and RITA-3) with 5-year outcomes [24]. Over five years, 14.7% (389 of 2,721) of patients randomly allocated to a routine invasive (RI) approach reach the endpoints cardiovascular death or non-fatal MI versus 17.9% (475 of 2,746) in the selective invasive (SI), more conservative strategy (HR: 0.81, 95% CI: 0.71–0.93; *p* = 0.002). The most marked treatment effect was on myocardial infarction (10.0% RI strategy vs. 12.9% SI strategy), and there were consistent trends for cardiovascular deaths (HR: 0.83, 95% CI: 0.68–1.01; *p* = 0.068) and all deaths (HR: 0.90, 95% CI: 0.77–1.05). 

There were 2.0% to 3.8% absolute reductions in cardiovascular death or MI in the low- and intermediate-risk groups and an 11.1% absolute risk reduction in highest-risk patients. 

Important in the interpretation of the trials is the revascularization rate in the conservative or selective invasive treatment arms. In the ICTUS trial, in the early invasive strategy as per protocol, 97% of patients underwent coronary angiography within 48 h and 98% during hospitalization. Coronary angiography during hospitalization was 96% in both early invasive groups of RITA-3 and FRISC-II, and by design, was to be performed <72 h from randomization and <7 days from admission for the index event, respectively. More importantly, coronary angiography during hospitalization in the non-invasive groups of FRISC-II and RITA-3 was only 7% and 16%, respectively, compared with 53% of patients in the selective invasive group in ICTUS. Therefore, selective invasive treatment might be an alternative strategy as long as ischemia detection is performed with a low threshold. This might explain the results of the ICTUS trial, in which the selective invasive treatment included ischemia detection before discharge. Furthermore, in the Netherlands, there is a high density of PCI and non-PCI angiography performing centers facilitating the transfer of unstable patients or patients with a positive ischemia detection test. 

In summary, a routine invasive strategy has benefit consisting of a reduction in recurrent myocardial infarction or death, with a larger benefit in higher-risk patients.

### 3.2. Timing of Invasive Treatment

The second decision involves the optimal timing of angiography following the routine invasive strategy. Different time windows are defined as immediate (within 2 h), early invasive (within 24 h) or delayed invasive (after 24 h but within 72 h). In the immediate strategy group no randomized clinical trials have been performed because of very-high-risk criteria identifying patients who need an urgent invasive approach. These criteria consist of: Hemodynamic instability or cardiogenic shock, refractory angina pectoris despite medical treatment, cardiac arrest or life-threatening arrhythmias, acute heart failure and recurrent dynamic ST- or T-wave changes. 

In patients with an out-of-hospital cardiac arrest (OHCA), but without ST-segment elevation on electrocardiography (ECG), the right timing of coronary angiography especially is still unknown [25,26]. Whether these patients should undergo immediate coronary angiography or delayed invasive treatment after neurological recovery was addressed in the Coronary Angiography after Cardiac Arrest (COACT) trial. In this multicenter randomized investigator-initiated trial 552 patients with NSTE-ACS and OHCA were randomly assigned to immediate angiography and delayed angiography [27]. The primary endpoint was survival at 90 days. At 90 days, no difference was seen in both groups (64.5% vs. 67.2%; OR, 0.89; 95% CI, 0.62–1.27; *p* = 0.51)

While a routine invasive approach in NSTE-ACS is recommended by clinical guidelines, the optimal timing of invasive strategy is still under debate. Multiple trials have compared immediate, early and delayed invasive treatment strategies, with discrepant results, and all were statistically underpowered to detect a mortality benefit. This issue was addressed in a meta-analysis of Jobs et al. which includes all randomized controlled trials comparing an early versus delayed invasive strategy and reporting all-cause mortality at least 30 days after in-hospital randomization [28]. The authors find that in eight trials (a total of 5324 patients) with a median follow-up of 180 days, no significant mortality reduction was seen in the early invasive group compared with the delayed invasive group (HR 0.81; 95% CI 0.64–1.03; *p* = 0.088). However, in a prespecified analysis of patients with elevated troponin, present diabetes, a GRACE risk score > 140 and older age (>75 years), lower mortality was seen with an early invasive strategy. Tests for interaction effects were inconclusive. We note that these meta-analyses are dominated by the largest TIMACS trial, providing the majority of patients for the meta-analysis. Furthermore, there is a large variety in the definition of early or delayed invasive strategy between the different trials. 

With regard to the GRACE risk score > 140, the randomized clinical TIMACS trial should be noted [29]. In this landmark trial, the largest RCT to date, 3031 patients with NSTE-ACS were randomly allocated to early invasive (within 24 h after randomization) or delayed intervention (≥36 h after randomization). At six months, no difference was seen in the primary outcome (a composite of death, myocardial infarction, or stroke) in the early-intervention group (9.6%) as compared to the delayed-intervention group (11.3%), HR in the early-intervention group, 0.85; 95% CI, 0.68–1.06; *p* = 0.15). The median time to angiography was 14 h in the early invasive strategy and 50 h in the delayed invasive group. Prespecified analysis of high-risk patients showed that early intervention improved the primary outcome in the third of patients who were at highest risk (i.e., GRACE risk score >140; HR, 0.65; 95% CI, 0.48–0.89). Albeit a positive subgroup in an overall negative trial, this finding was recently reproduced. In the VERDICT trial (Very Early Versus Deferred Invasive Evaluation Using Computerized Tomography) 2147 patients with a clinical suspicion of NSTE-ACS were 1:1 randomized to very early invasive coronary angiography (within 12 h of diagnosis) or standard invasive (within 48–72 h) [30]. The primary endpoint was a combination of all-cause death, non-fatal recurrent myocardial infarction, hospital admission for refractory myocardial ischemia, or hospital admission for heart failure. With a median follow-up time of 4.3 years, no difference was seen in the primary endpoint in the very early group (27.5%) versus the standard group (29.5%), (HR, 0.92; 95% CI, 0.78–1.08. However, the benefit of an early invasive treatment strategy for predefined high-risk patients (GRACE risk score > 140) was also confirmed in this trial (HR, 0.81; 95% CI, 0.67–1.01; *p* value for interaction = 0.023). 

### 3.3. Current Guideline Recommendations

As outlined in Table 1, the ESC recommends CAG within two hours in patients with very-high-risk criteria. In patients with a rise and/or fall in cardiac troponin, dynamic ST-or T-wave changes and a GRACE risk score > 140, the guidelines recommend an early invasive strategy. Intermediate risk criteria mandate a delayed invasive approach (within 72 h).

### 3.4. Summary and Clinical Perspective

In Table 2, the key points in antithrombotic treatment and invasive approach in NSTE-ACS patients are outlined.

With the introduction of clopidogrel and the potent P2Y_12_ inhibitors ticagrelor and prasugrel, patients benefit from significant reductions in cardiovascular mortality and recurrent myocardial infarction when added to aspirin, the cornerstone of ACS treatment. Strategies in combining antithrombotic treatments will be the focus in future research where ischemic benefit should outweigh the risk of (major) bleeding. While waiting for these trials, an invasive evaluation of coronary anatomy should be a mandatory part of the treatment strategy in high-risk patients presenting with NSTE-ACS.

## Figures and Tables

**Table 1 jcm-09-02578-t001:** Risk criteria mandating invasive strategy in non-ST segment elevation acute coronary syndrome (NSTE-ACS). Adapted from the European Society of Cardiology (ESC) guidelines [8].

**Very-High-Risk Criteria**
Hemodynamic instability or cardiogenic shock
Recurrent or ongoing chest pain refractory to medical treatment
Life-threatening arrhythmias or cardiac arrest
Mechanical complications of MI
Acute heart failure
Recurrent dynamic ST-T wave changes, particularly with intermittent ST-elevation
**High-Risk Criteria**
Rise or fall in cardiac troponin compatible with MI
Dynamic ST-or T-wave changes (symptomatic or silent)
GRACE score > 140
**Intermediate-Risk Criteria**
Diabetes mellitus
Renal insufficiency (eGFR < 60 mL/min/1.73 m^2^
LVEF < 40% or congestive heart failure
Early post-infarction angina
Prior PCI
Prior CABG
GRACE risk score > 109 and <140
**Low-Risk Criteria**
Any characteristics not mentioned above

CABG = coronary artery bypass graft; eGFR = estimated glomerular filtration rate; GRACE = Global Registry of Acute Coronary Events; LVEF = left ventricular ejection fraction; PCI= percutaneo coronary intervention; MI = myocardial infarction.

**Table 2 jcm-09-02578-t002:** Key points in antithrombotic pretreatment and invasive strategies in NSTE-ACS.

**Key Points in Antiplatelet Therapy**
Aspirin still remains the cornerstone
DAPT with a potent P2Y_12_ inhibitor is recommended
Pretreatment with P2Y_12_ inhibitors have no benefits on MACE
**Key Points in Anticoagulant Therapy**
Anticoagulation should be started at diagnosis
UFH is the first-line therapy
Fondaparinux is the first-line therapy with a selective invasive or conservative management
**Key Points in Invasive Coronary Angiography and Revascularization**
Routine invasive strategy is recommended by clinical guidelines
An early invasive strategy in patients with high-risk criteria is recommended

DAPT = dual antiplatelet therapy; MACE = major adverse cardiovascular events (a composite endpoint of cardiovascular death, recurrent myocardial infarction, stroke, urgent revascularization); UFH = unfractionated heparin.

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
