# Peer review of "Antithrombotic PreTreatment and Invasive Strategies in Patients with Non-ST-Segment Elevation Acute Coronary Syndrome"

_jcm, 2020, doi:10.3390/jcm9082578_

Round 1
Reviewer 1 Report
The presented manuscript covers well the oral antithrombotic and invasive management of patients with NSTE-ACS. It is quite a short article with a single table and I wonder if there is also room to discuss the use of parenteral anticoagulants (fondaparinux, LMW heparin, unfractionated heparin or bivalirudin) at the time of diagnosis as I believe these fall in the remit of this review. Similarly, if there is space there could be a brief discussion of the antithrombotic management of those with NSTE-ACS and an indication for oral anticoagulation, as this is a topic that causes clinicians some confusion.
There are just a few other minor tweaks to the article that I think would improve it, listed below. Line numbers refer to those provided on the pdf proof.
Line 33 – perhaps harmonise the use of ‘thrombocyte’ and ‘platelet’ to refer just to platelets.
Lines 30 and 37 – there are two uses of the word ‘cornerstone’ close together and it might read better to change one of these.
Line 44 – perhaps briefly mention that TRILOGY ACS failed to show any benefit of prasugrel over clopidogrel in those with NSTE-ACS not undergoing revascularisation (I realise this is alluded to in the ‘recommendations’ section).
Line 117 – ‘off’ should be ‘of’.
Line 118 – there is a line break that should be removed.
Line 155 – there is an additional comma after ‘PCI’ that should be removed.
Line 158 – I believe ‘are’ should be removed.
Line 203 and 256 – a full stop is needed at the end of the sentence.
Author Response
Point 1: is there room to discuss the use of parenteral anticoagulants (fondaparinux, LMW heparin, unfractionated heparin or bivalirudin) at the time of diagnosis
Response 1: we agree with the reviewer that these parenteral agents (including Gp2b/3a) should be addressed in this review. We add important literature regarding GPp 2b/3a inhibitors (line 132-144) and rewrote the anticoagulants paragraph focusing on LMW heparin, unfractionated herapin, fondaparinux and bivalirudin. Line 151 - 197
Point 2: there could be a brief discussion of the antithrombotic management of those with NSTE-ACS and an indication for oral anticoagulation, as this is a topic that causes clinicians some confusion.
Response 2: important comment. Our review is part of different articles focusing on the antithrombotic treatment in ACS. The management of oral anticoagulation (e.g. atrial fibrillation), in patients with ACS will be addressed in another review article. In our review focus on the treatment in the acute phase of NSTE-ACS.
Point 3: Line 33 – perhaps harmonise the use of ‘thrombocyte’ and ‘platelet’ to refer just to platelets.
Response 3: good suggestion. We changed the reviewer suggestion in the text (line 33,thrombocyte has been changed to platelet)
Point 4: Lines 30 and 37 – there are two uses of the word ‘cornerstone’ close together and it might read better to change one of these.
Response 4: We changed cornerstone in “basis” (line 37)
Point 5: Line 44 – perhaps briefly mention that TRILOGY ACS failed to show any benefit of prasugrel over clopidogrel in those with NSTE-ACS not undergoing revascularisation (I realise this is alluded to in the ‘recommendations’ section).
Response 5: in this review we focus on the antithrombotic management of NSTE-ACS patients undergoing coronary angiography and/or subsequent PCI. Nowadays, CAG is almost mandatory in patients with NSTE-ACS. However whether this should be routine invasively or selective invasive is a matter of debate
Point 6: Line 117 – ‘off’ should be ‘of’.
Response 6: we rewrote this section. The focus will be on anticoagulants in the acute phase of ACS: UFH, LMWH, bivalirudin. See also response 1
Point 7: Line 118 – there is a line break that should be removed.
Response 7: we rewrote this section (see response 1 and 6)
Point 8: Line 155 – there is an additional comma after ‘PCI’ that should be removed.
Response 8: we removed the additional comma (line 242)
Point 9: Line 158 – I believe ‘are’ should be removed.
Response 9: we removed “are” (line 245)
Point 10: Line 203 and 256 – a full stop is needed at the end of the sentence.
Response 10: we changed this in the tekst (line 364 and 434)
Reviewer 2 Report
The abbreviation for hazard ratio -- HR -- throughout the paper. Define abbreviation then use it consistently. Same for OR. There are numbers that don't map to a meaning; there are just numbers with a bunch of punctuation. There is way too much text; there should be a couple of tables would help. There are other things that are marked in the paper itself.

Author Response
We thank the reviewer for his comments. The comments the reviewer made was based on an earlier version of our manuscript. Based on earlier reviewer comments, we did major revisions (including re writing the anticoagulant section in the acute phase of ACS). Below you'll find the point-by-point response:
POINT 1: abbreviation for hazard ratio and odds ratio should be consistently
RESPONSE 1: we abbreviated hazard ratio (HR) and odds ratio (OR) consistently in our manuscript
POINT 2: There are numbers that don't map to a meaning there are just numbers with a bunch of punctuation
RESPONSE 2: we correct the punctuation in the text
POINT 3: much text, couple of tables
RESPONSE 3: This is a good point and also pointed by another reviewer. Therefore we added an extra table (table 2) to give a concise summary of the most important key points
POINT 4: marked comments in the paper
RESPONSE 4: we looked at the marked comments and revise the text as advised by the reviewer
Reviewer 3 Report
This review focused on anti-thrombotic treatment strategies (before or at the time of coronary angiography) and the invasive management in non-ST segment elevation acute coronary syndrome (NSTE-ACS).
This is a clinically relevant and actual review. The strengths of this review provide us the update on the anti-thrombotic treatment of patients with NSTE-ACS in the current era.
Author Response
We thank the reviewer for his comments
Reviewer 4 Report
This is a comprehensive summary of the current landscape of antithrombotic therapy for NSTEACS. Although well written, the review would benefit from greater perspective regarding the limitations of current antithrombotic strategies (ie bleeding or limited antiplatelet effect) and should consider novel therapies in development in this space (Chan and Weitz, Circ Res, 2019; McFadyen et al, Nature Reviews Cardiology 2018). Likewise, some discussion of antithrombotic strategies in patients with AF and NSTEACS. The review seems to end fairly abruptly some perspective and summary here would round the review out nicely.
Additionally, the text is relatively dense, and some of the trials discussed in the text could be presented/summarised in a table(s) for the reader to reference and to break up the text.
Author Response
POINT 1: the review would benefit from greater perspective regarding the limitations of current antithrombotic strategies (ie bleeding or limited antiplatelet effect) and should consider novel therapies in development in this space (Chan and Weitz, Circ Res, 2019; McFadyen et al, Nature Reviews Cardiology 2018).
RESPONSE 1: this is an important field of interest. Our review is part of a special issue on ACS in the Journal of Clinical Medicine and the topics mentioned by the reviewer will be addressed in another review article.
POINT 2: Likewise, some discussion of antithrombotic strategies in patients with AF and NSTEACS.
RESPONSE 2: this is an important field of interest. Our review is part of a special issue on ACS in the Journal of Clinical Medicine and the topics mentioned by the reviewer will be addressed in another review article.
POINT 3: The review seems to end fairly abruptly some perspective and summary here would round the review out nicely.
RESPONSE 3: we agree with the reviewer. In our review article a concise summary has been added at the end of our manuscript in line 445-451
POINT 4: Additionally, the text is relatively dense, and some of the trials discussed in the text could be presented/summarised in a table(s) for the reader to reference and to break up the text.
RESPONSE 4: we agree with the reviewer that a table will break up the text. We choose to solve this issue to insert a table with the most important key points regarding antithrombotic and invasive treatment in ACS. Table 2 has been added in line 453 in the revised manuscript
Round 2
Reviewer 4 Report
I am satisfied with the author's comments.
Author Response
We thank the reviewer for reading our manuscript.
This manuscript is a resubmission of an earlier submission. The following is a list of the peer review reports and author responses from that submission.